# Analysis on Bearing Behavior of Single Pile under Combined Action of Vertical Load and Torque in Expansive Soil

**Zhi Wang [1], Jie Jiang [2],*, Shunwei Wang [2], Chenzhi Fu [2] and Di Yang [1]**

[1]  China Railway Construction Group Co., Ltd., Beijing 100040, China
[2]  School of Civil Engineering and Architecture, Guangxi University, Nanning 530004, China
*   Correspondence: jiejiang@gxu.edu.cn

**Abstract:** To reasonably analyze the bearing characteristics of curved beam bridge pile foundation under the combined action of vertical force ($V$) and torque ($T$) in expansive soil, a method of determining $T_u$ (limit torque) under the action of $V$ is proposed. Considering the effects of expansive force and ground heave after immersion, the load transfer function at the pile–soil interface with positive and negative friction resistance is established. The nonlinear solution of a single pile under vertical load is achieved by the finite difference method. Subsequently, the circumferential limit friction resistance is modified, and a loop iteration program is compiled through MATLAB. Thus, the bearing capacity of single pile under the loading path of $V{\rightarrow}T$ is obtained. The corresponding failure envelope curve is drawn and verified by laboratory model tests. Based on the verified solution, the bearing capacity envelope curve and deformation characteristics of single pile are analyzed. The influence of the expansion rate on the bearing capacity envelope curve is discussed, and reasonable engineering suggestions are put forward.

**Keywords:** single pile; expansive soil; bearing characteristics; $V{\rightarrow}T$ loading path; model test

## 1. Introduction

Expansive soil is a highly plastic and cohesive soil, typically existing in a hard and strong state in its natural state. It is sensitive to moisture and can swell when it absorbs water, resulting in significant deformation when it is in a free state. If deformation is restricted, it can generate expansive force. Conversely, when it loses water, its volume can shrink, leading to cracks and a decrease in strength. Its high degree of expansiveness can cause building structures to tilt, deform, and undergo uneven settlement, presenting significant engineering problems. Expansive soil is widely distributed in more than 60 countries in the world, and China is also one of the major countries [1]. Pile foundations are often used in practical engineering to reduce the impact of expansion and contraction on the superstructure. Curved beam bridges are widely used in constructing urban overpass ramp bridges and elevated urban expressways to better adapt to terrain constraints. When prestressing the curved girder bridge during construction, the pile foundation is first subjected to a vertical load ($V$) and then to torque ($T$), which corresponds to the $V{\rightarrow}T$ loading path, and the stress condition of the pile foundation in expansive soil is complicated. The Technical Code for Buildings in Expansive Soil Regions [2] does not give a calculation method for the bearing capacity of a single pile under vertical or torque loads.

Moreover, the Technical Code for Building Pile Foundations [3] proposes that the unidirectional force of the single loaded piles should be calculated separately and then superimposed for the combined loading pile. However, there is a complex coupling effect between $V$ and $T$ [4,5]. According to the method given in the code, the bearing capacity of a single pile under a $V$–$T$ combined load is not safe, so it is necessary to conduct in-depth research.

Scholars have studied the axial force of a single pile in expansive soil. In terms of experiments, Williams and Donaldson [6], Wang et al. [7], and Fan [8] proved that

immersion in expansive soil would raise the single pile and generate an axial internal force in the pile. In theory, Poulos and Davis [9] considered the influence of the expansion force, while the internal force of the single pile was solved by the elastic theory method after immersion. However, there is a significant error between the calculated results and the experimental data. Soundara and Robinson [10] proposed a hyperbolic model to predict the uplift force of single pile in expansive soil based on the model pile uplift test, pile–soil interface shear test, and consolidation test. Nonetheless, test process is cumbersome. Zhang et al. [11] proposed the analytical solution of internal force and displacement for the single pile after immersion, considering the variety of soil expansion rates and stiffnesses with depth. The drawback is that calculation parameters are difficult to determine. For the vertically loaded piles in expansive soil, Xiao et al. [12] used the shear displacement method and superposition principle to solve the analytical solution. There were errors in the calculation results still. To solve the displacement response of a single pile in expansive unsaturated soil under vertical load, Liu and Vanapalli [13] considered the influence of matric suction on the ultimate frictional resistance of the pile-side. It should be known that obtaining the water–soil characteristic curve takes much time. In conclusion, the current theories for determining the axial force and the displacement response under vertical load in expansive soil after immersion are not perfect.

For *V–T* combined loaded piles, Georgiadis et al. [14,15] used independent nonlinear axial and torsional springs to simulate the interaction of the pile–soil interface, and the numerical solution of pile top load-displacement was obtained in a clay foundation. Meng and Fan [16] analyzed the difference in the bearing capacity envelopes of the *V–T* combined loaded piles under different loading paths using three-dimensional finite element software in clay foundation. Jiang et al. [17] obtained the nonlinear solution of *V–T* combined loaded piles in clay by finite difference. Zou et al. [5] obtained the elastoplastic analytical solution of *V–T* combined loaded piles in the Gibson foundation. However, the above studies were carried out in non-expansive soil. Due to axial force and displacement of the pile after immersion in expansive soil, the mechanical mechanism of *V–T* combined loaded piles in expansive soil foundations is significantly different from the above studies. Therefore, the above theories are not fully applicable to expansive soil foundations.

In summary, there are few reports about *V–T* combined loaded piles in expansive soil foundations. Because scholars have not perfected the research on the internal force and displacement of the pile under the vertical load after immersion, it is impossible to determine the bearing capacity of a single pile under *V–T* combined loads. A nonlinear solution of the internal force and displacement of a single pile under vertical load is proposed in this paper. On this basis, the circumferential limit friction of the pile under torque loading is corrected. The numerical solution of the load-displacement of the single pile under the *V→T* loading path is obtained, which is verified by experiments. Subsequently, the bearing characteristics of a single pile under the *V→T* loading path in an expansive soil foundation are analyzed, and the corresponding engineering measures are put forward.

## 2. Theoretical Derivation

### 2.1. The Ultimate Friction Resistance of the Pile Side

The expansion force will be generated after immersion in the expansive soil. The lateral earth pressure is composed of the static earth pressure and the lateral swelling force. According to the research of Nelson et al. [18], the lateral pressure of the soil can be expressed as:

$$\sigma = \begin{cases} P_a + \beta\sigma_{cv} \leq P_a + P_p & (0 \leq z \leq h_0) \\ P_a & (z > h_0) \end{cases} \tag{1}$$

where $\sigma$ is the lateral earth pressure of the expansive soil after immersion; $h_0$ is the atmospheric influence depth, which can be determined according to specification [2]. The coefficient of lateral pressure of the expansive force $\beta$ is equal to 0.7~1; $\sigma_{cv}$ is the expansive force, and it can be measured in the laboratory test; $P_a$ is the static earth pressure and $P_p$ is the passive earth pressure.

The ultimate frictional resistance of the pile–soil interface is:

$$\tau_f(z) = \sigma \tan\varphi' + c \tag{2}$$

where $c$ and $\varphi'$ are the cohesive force and the effective internal friction angle of the pile–soil interface, respectively. $\varphi'$ is equal to 0.6~0.9$\varphi$ according to the different soil properties [19], and $\varphi$ is the internal friction angle of the soil.

### 2.2. Solution of Piles under Single Vertical Load

The solution of vertically loaded piles in expansive soil foundations can be considered in two stages: in the first stage, the pile displacement caused by soil expansion is calculated, and in the second stage, the displacement of the pile is regarded as the known condition to calculate the displacement response of a single pile under vertical load.

### 2.2.1. Pile Displacement Caused by Soil Expansion

The expansive soil foundation will be uplifted under the action of immersion or rainfall. The uplift height of expansive soil is linearly with depth within the range of atmospheric influence [8], and the soil uplift is equal to 0 below the atmospheric influence depth range. Thus, the displacement of the foundation:

$$w_s(z) = \begin{cases} w_{s0} - \dfrac{w_{s0}}{h_0}z & (0 \leq z \leq h_0) \\ 0 & (z > h_0) \end{cases} \tag{3}$$

where $w_s(z)$ is the uplift height of expansive soil at depth $z$; and $w_{s0}$ is the uplift height of the expansive soil surface; $h_0$ is the depth at which expansion occurs, and soil below the depth of $h$ will not experience expansion.

The pile–soil interface model is shown in Figure 1, when the expansive soil is immersed in water. The vertical positive and negative frictional resistances of the pile–soil interface conform to the load transfer function proposed by Kraft et al. [20]:

$$\tau_v(z) = \begin{cases} \dfrac{G_s s_v(z)}{r_0 \ln(r_m/r_0 - \psi/1 - \psi)} & (s_v(z) \geq 0) \\ -\dfrac{G_s |s_v(z)|}{r_0 \ln(r_m/r_0 - \psi/1 - \psi)} & (s_v(z) < 0) \end{cases} \tag{4}$$

where $s_v(z) = w(z) - w_s(z)$ is the vertical pile–soil relative displacement; and $G_s$ is the initial shear modulus of the soil under a small strain; $\psi = \tau_v(z)R_f/\tau_f(z)$; and $R_f$ is the fitting constant of stress–strain curve, which is equal to 0.9~1.0; $r_0$ is the pile section radius; and $r_m$ is the effective influence radius.

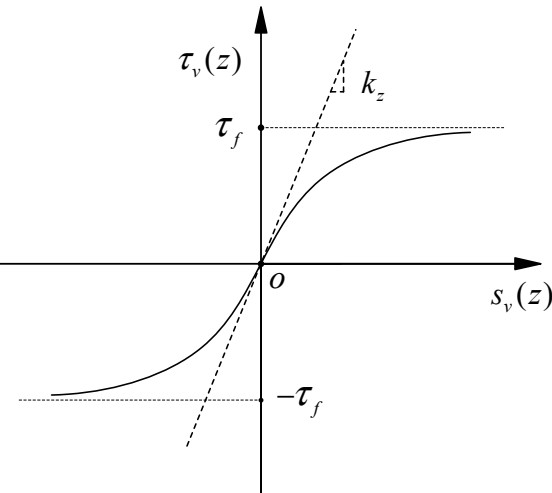

**Figure 1.** Pile–soil interface model of expansive soil after immersion.

According to the equilibrium conditions of pile microelement, we can obtain:

$$\frac{dP(z)}{dz} = -U_p \tau_v(z) \tag{5}$$

where $U_p$ is the perimeter of the pile section; $P(z)$ and $\tau_v(z)$ are axial force and vertical shaft resistance of the pile at depth $z$, respectively.

The elastic compression of the pile microelement can be written as:

$$dw_{swell}(z) = -\frac{P(z)dz}{E_P A_p} \tag{6}$$

where $w_{swell}(z)$ is the displacement of the pile top caused by the uplift of expansive soil; $E_P$ and $A_p$ are the elastic modulus and cross-sectional area of the pile, respectively.

The pile control equation can be obtained through simultaneous Equations (5) and (6):

$$\frac{d^2 w_{swell}(z)}{dz^2} - \frac{U_p}{E_p A_p} \tau_v(z) = 0 \tag{7}$$

The tangent stiffness of the soil on the pile-side can be expressed as:

$$k_v = \frac{U_p \tau_v(z)}{s_v(z)} \tag{8}$$

Given $\lambda = \sqrt{\frac{k_v}{E_p A_p}}$, the pile control equation can be written as:

$$\frac{d^2 w_{swell}(z)}{dz^2} - \lambda^2 (w_{swell}(z) - w_s(z)) = 0 \tag{9}$$

The pile top and tip boundary conditions are as follows:

➢  For the pile top, axial force can be written as:

$$P(0) = 0 \tag{10}$$

➢  For the pile tip, Wang et al. [7] proved that the pile tip and soil are separated after immersion through experiments. Hence, the force at the pile tip is expressed as follows:

$$P(L) = 0 \tag{11}$$

As shown in Figure 2, the central difference method is used to discretize the pile length into $n$ equal elements, and a virtual equal division node is added to the pile top and pile tip. Substituting the boundary conditions of the pile tip and pile top into Equation (9), we can obtain the system of equations:

$$\left[K_v{'}\right]\{w_{swell}\} = \{F_v{'}\} \tag{12}$$

where $\{w_{swell}\}$ is the vertical displacement vector of the pile node,

$$\{w_{swell}\} = \{w_{swell,0} w_{swell,1} \cdots w_{swell,i} \cdots w_{swell,n-1} w_{swell,n}\}^T \tag{13}$$

$\{F_v{'}\}$ is the vertical load vector of the pile node,

$$\{F_v{'}\} = \left\{-\lambda_0{}^2 h^2 w_{s,0} - \lambda_1{}^2 h^2 w_{s,1} \cdots - \lambda_i{}^2 h^2 w_{s,i} \cdots - \lambda_{n-1}{}^2 h^2 w_{s,n-1} - \lambda_n{}^2 h^2 w_{s,n}\right\}^T \tag{14}$$

$[K_v']$ is the vertical stiffness matrix of the pile and it is shown as follows:

$$[K_v'] = \begin{bmatrix} A_0 & 2 & & & & & \\ 1 & A_1 & 1 & & & & \\ & \ddots & \ddots & \ddots & & & \\ & & 1 & A_i & 1 & & \\ & & & \ddots & \ddots & \ddots & \\ & & & & 1 & A_{n-1} & 1 \\ & & & & & 2 & A_n \end{bmatrix}_{(n+1)\times(n+1)} \tag{15}$$

where $A_i = -(\lambda_i^2 h^2 + 2)$.

Solving Equation (12), the vertical displacement along the pile can be obtained as

$$\{w_{swell}\} = [K_v']^{-1}\{F_v'\} \tag{16}$$

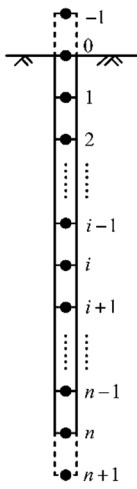

**Figure 2.** Schematic diagram of differential dispersion of single pile after immersion.

The iterative process for solving Equation (12) using MATLAB is as follows:

(i) Assign the vertical displacement vector of pile node $\{w_{swell}\}$ to zero vector. The relative displacement of pile and soil is calculated. Then, the vertical stiffness matrix $[K_v']$ of the pile is obtained.

(ii) Calculating displacement vector $\{w_{swell}\}$ of pile node by Equation (16).

(iii) A new vertical stiffness matrix $[K_v']^k$ is obtained using the new displacement vector $\{w_{swell}\}^k$, and a new node displacement $\{w_{swell}\}^{k+1}$ is obtained using the new vertical stiffness matrix.

(iv) Taking the maximum value of elements in the $\left|\{w_{swell}\}^{k+1} - \{w_{swell}\}^k\right|$ as iterative control error. If the error is greater than the limit value, repeat (ii)~(iv) until the iterative error is less than the limit value.

### 2.2.2. Nonlinear Solution of Single Pile under Vertical Load

The expansive soil would swell after immersion. The soil around the pile exerts an uplift force above the pile's neutral point (i.e., the vertical friction resistance equals 0), and it exerts a pulldown force below the neutral point. When a vertical load is applied to the pile top, the shear direction remains unchanged above the neutral point. However, the direction of relative movement between the pile and soil is opposite to that during immersion below the neutral point, and reverse shear will occur at the pile–soil interface.

Reverse shear can be considered as an unloading process. Alonso et al. [21] and Comodromosa et al. [22] believed that the unloading stiffness equals the initial tangent stiffness during the unloading process. Therefore, the pile–soil interface model in Figure 1 can be improved. As shown in Figure 3, it can uniformly be expressed the load transfer model of pile–soil under vertical loading after immersion. *OA* and *OB* are the immersion swelling stages, respectively; *AC* is the continuous loading section; *BD* is the unloading section; and *DE* is the reverse loading section. For the convenience of description, the pile–soil interface load transfer model can be expressed as:

$$\tau(z) = \begin{cases} \dfrac{G_s s(z)}{r_0 ln(r_m/r_0 - \psi/1-\psi)} & OA, AC \\[2mm] -\dfrac{G_s |s(z)|}{r_0 ln(r_m/r_0 - \psi/1-\psi)} & OB \\[2mm] k_z\left(s(z) - s_p\right) & BD \\[2mm] \dfrac{G_s\left(s(z) + |s_p(z)|\right)}{r_0 ln(r_m/r_0 - \psi/1-\psi)} & DE \end{cases} \tag{17}$$

where $w_{load}$ is a new displacement of pile under vertical load after immersion; Here, vertical pile–soil relative displacement $s(z) = w_{swell} + w_{load} - w_s(z)$; and $s_p(z)$ is residual displacement of pile–soil interface.

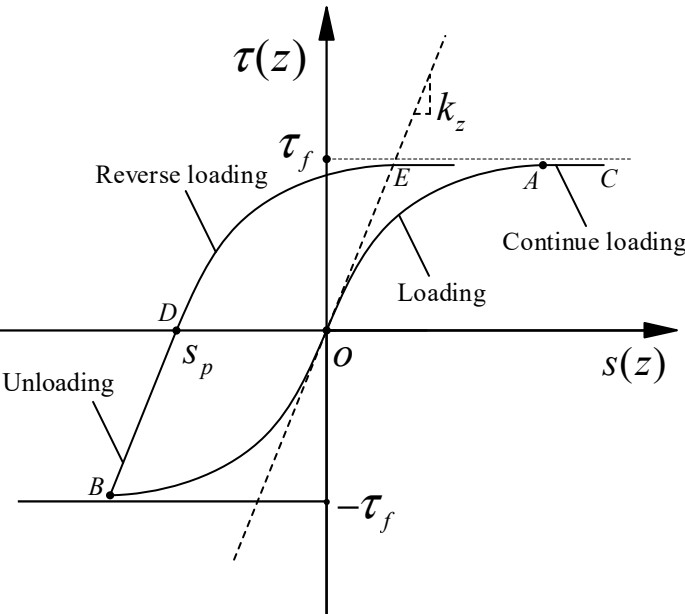

**Figure 3.** Load transfer curve of pile–soil interface under vertical load.

The pile control equation can also be expressed as:

$$\frac{d^2 w(z)}{dz^2} - \lambda'^2 s(z) = 0 \tag{18}$$

The boundary conditions of the pile top and pile tip are as follows:
➢  For the pile top:
$$w_{load}(0) = w_0 \tag{19}$$

where $w_0$ is the known reload displacement.

➢  For the pile tip, the vertical bearing capacity can be expressed as:

$$P(l) = \begin{cases} \dfrac{s_b(l)}{1/K_{bz} + s_b(l)/q_{ult}} & (s_b(l) \geq 0) \\[2mm] 0 & (s_b(l) < 0) \end{cases} \tag{20}$$

where $s_b(l)$ is the vertical pile–soil relative displacement at the pile tip, which is calculated by the following formula:

$$s_b(l) = w_{swell}(l) + w_{load}(l) - w_s(l) \tag{21}$$

Randolph et al. [23] and Mylonakis et al. [24] proposed the empirical formula of $K_{bz}$ (i.e., the pile-tip soil initial stiffness) as follows:

$$K_{bz} = \frac{4G_s r_0}{1 - v_s} \tag{22}$$

where $v_s$ is the Poisson's ratio of the soil.

The tangent stiffness of the pile tip soil is

$$K_{bz}' = \begin{cases} \frac{1}{1/K_{bz} + s_b(l)/q_{ult}} & (s_b(l) \geq 0) \\ 0 & (s_b(l) < 0) \end{cases} \tag{23}$$

For saturated viscous soils with poor drainage conditions, the ultimate pile tip resistance $q_{ult}$ can be expressed as follows [25]:

$$q_{ult} = 5.14c + q \tag{24}$$

where $q$ is the average vertical pressure on the side of the pile tip plane, and it can be calculated by the following formula:

$$q = \frac{(1 + 2k_0)\gamma L}{3} \tag{25}$$

where $k_0$ is the static earth pressure coefficient; and $\gamma$ is the soil gravity.

As shown in Figure 4, the central difference method is used to discretize the pile length into $n$ equal elements, and a virtual equal division node is added to the pile tip. Substituting the boundary conditions of the pile tip and top into Equation (18), we can obtain the system of equations:

$$[K_z''] \{w_{load}\} = \{F_z''\} \tag{26}$$

where $\{w_{load}\}$ is the vertical displacement vectors of the pile nodes under vertical load, $\{w_{load}\} = \{w_{load,0} w_{load,1} \cdots w_{load,i} \cdots w_{load,n-1} w_{load,n}\}^T$; $[K_z'']$ is the pile stiffness matrix under vertical load; $\{F_z''\}$ is vertical load vector of the pile nodes.

$$[K_z''] = \begin{bmatrix} -2 & 1 & & & & & \\ 1 & -2 & 1 & & & & \\ & \ddots & \ddots & \ddots & & & \\ & & 1 & -2 & 1 & & \\ & & & \ddots & \ddots & \ddots & \\ & & & & 1 & -2 & 1 \\ & & & & & 2 & -2 - \frac{2hK_{bz}'}{E_p A_p} \end{bmatrix} \tag{27}$$

$$\{F_z''\} = \begin{Bmatrix} \lambda_1'^2 h^2 s_1 - w_{swell,0} + 2w_{swell,1} - w_{swell,2} - w_0 \\ \lambda_2'^2 h^2 s_2 - w_{swell,1} + 2w_{swell,2} - w_{swell,3} \\ \vdots \\ \lambda_i'^2 h^2 s_i - w_{swell,i-1} + 2w_{swell,i} - w_{swell,i+1} \\ \vdots \\ \lambda_{n-1}'^2 h^2 s_{n-1} - w_{swell,n-2} + 2w_{swell,n-1} - w_{swell,n} \\ \lambda_n'^2 h^2 s_n - 2w_{swell,n-1} + \left(2 + \frac{2hK_{bz}'}{E_p A_p}\right) w_{swell,n} - \frac{2hK_{bz}'}{E_p A_p} w_{s,n} \end{Bmatrix} \tag{28}$$

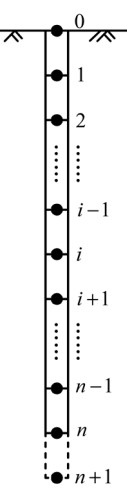

**Figure 4.** Schematic diagram of differential dispersion of single pile under vertical load after water immersion.

Solving Equation (26) can obtain the vertical displacement of a single pile along the pile under vertical load as follows:

$$\{w_{load}\} = [K_z'']^{-1}\{F_z''\} \tag{29}$$

The nonlinear solution process of the vertically loaded pile considering the influence of immersion in expansive soil is as follows:

To calculate the displacement of a vertically loaded pile considering the influence of immersed expansive soil, the iterative process calculated by MATLAB is as follows:

(i)   Calculating the soil displacement $w_s(z)$ around the pile after immersion and the vertical displacement vector $\{w_{swell}\}$ of the pile caused by the uplift of the soil, then the pile–soil relative displacement is achieved.

(ii)  Solving the stiffness matrix $[K_z'']$ of the pile under vertical loading, then the vertical displacement vectors $\{w_{load}\}^k$ of the pile nodes under vertical load can be calculated by the Equation (29).

(iii) Based on the vertical displacement vectors $\{w_{load}\}^k$, repeating steps (ii), then the vertical displacements update to be $\{w_{load}\}^{k+1}$.

(iv)  Taking the maximum value of elements in $\left| \{w_{load}\}^{k+1} - \{w_{load}\}^k \right|$ as iterative control error. If the error is greater than the limit value, repeat (ii)~(iv) until the iterative error is less than the limit value.

Substituting the relative displacement of the pile and soil obtained by the above numerical solution into Equation (17), the vertical friction resistance of the element $\tau_{sv}(i)$ can be obtained.

Vertical load on pile top can be obtained by forward difference format as follows:

$$P_0 = E_p A_p \frac{(w_{swell,1} + w_{load,1}) - (w_{swell,0} + w_{load,0})}{h} \tag{30}$$

*2.3. Displacement Solution of Single Piles under V→T Loading Path*

2.3.1. Determination of Circumferential Ultimate Frictional Resistance

The piles will rise after the expansive soil is soaked. The stress condition on the pile–soil interface is complicated under *V–T* combined loads. Both soil expansion and vertical load will induce vertical friction, and torque will induce circumferential friction. The pile can be divided into *n* equal elements. The composite friction resistance of each element $\tau(i)$ can be decomposed into vertical friction resistance $\tau_{sv}(i)$ and circumferential

friction resistance $\tau_t(i)$ since both vertical load and torque will induce shear effects at the pile–soil interface, as shown in Figure 5a. Hence, it is assumed that the pile–soil interface friction meets the following conditions (Figure 5b):

$$\tau(i) = \sqrt{[\tau_{sv}(i)]^2 + [\tau_t(i)]^2} \le \tau_f(i) \tag{31}$$

where $\tau_f(i)$ is pile–soil interface ultimate friction resistance of element $i$.

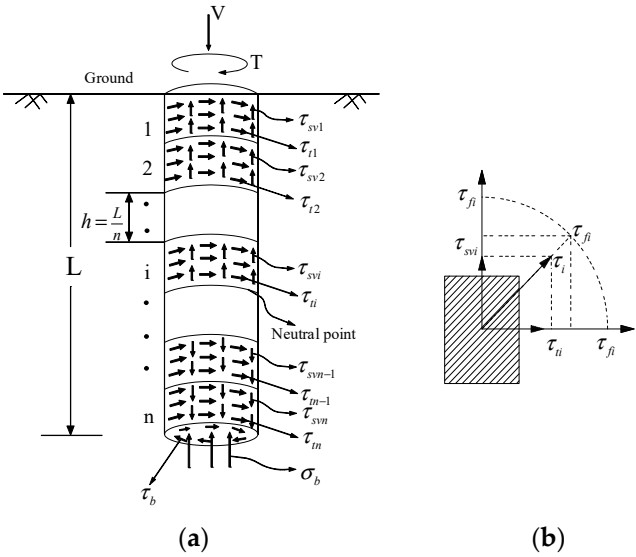

**Figure 5.** (**a**) Discrete model of pile surface (**b**) Shear element model.

The circumferential ultimate frictional resistance of the pile side decreases after applying vertical load. The following formula can obtain the circumferential ultimate frictional resistance in the $V \rightarrow T$ loading path:

$$\tau_{tf}(i) = \sqrt{\left[\tau_f(i)\right]^2 - [\tau_{sv}(i)]^2} \tag{32}$$

where $\tau_{tf}(i)$ is the ultimate circumferential friction resistance of the pile-side.

2.3.2. Displacement Solution of Single Pile under $V \rightarrow T$ Loading Path

The Jiang et al. [17] gives the calculation method of a single pile under torsion loading. The load transfer function of the pile–soil interface still adopts the method proposed by Kraft et al. [20].

$$\tau_t(z) = \frac{G_s s_t(z)}{r_0 ln(r_m/r_0 - \psi/1 - \psi)} \tag{33}$$

where $\tau_t(z)$ is the circumferential frictional resistance of the pile side; $s_t(z)$ is the relative torsional displacement of the pile–soil at depth $z$, $s_t(z) = r_0\theta(z)$, Note that in the equation, $\psi = \tau_t(z)R_f/\tau_{tf}(z)$.

The displacement control equation of a single pile under torque loading is [17]:

$$\frac{d\theta^2(z)}{dz^2} = \alpha^2\theta(z) \tag{34}$$

where $\alpha = \sqrt{\frac{k_\theta \pi D^3}{4G_p J_p}}$; and $k_\theta(z) = \frac{\tau_t(z)}{s_t(z)}$.

Using the finite difference method, the Equation (34) can be rewritten as:

$$\left[K_t{}'\right]\{\theta\} = \{T_t{}'\} \tag{35}$$

where $\{\theta\}$ is pile node torsion angle vector, $\{\theta\} = \{\theta_0 \theta_1 \dots \theta_i \dots \theta_{n-1} \theta_n\}^T$; and $\{T_t'\}$ is the torque load vector, $\{T_t'\} = \left\{-2Th/\left(G_p J_p\right) 0 \dots 0 \dots 0 - 3T_b h/\left(8G_L r_0^3\right)\right\}^T$; $[K_t']$ is torque stiffness matrix,

$$[K_t'] = \begin{bmatrix} B_0 & 2 & & & & & \\ 1 & B_1 & 1 & & & & \\ & \ddots & \ddots & \ddots & & & \\ & & 1 & B_i & 1 & & \\ & & & \ddots & \ddots & \ddots & \\ & & & & 1 & B_{n-1} & 1 \\ & & & & & & B_n \end{bmatrix} \tag{36}$$

where $T$ is the torque at the pile top; $G_p$ is the shear modulus of the pile; $J_p$ is the polar moment of inertia of pile section; $G_L$ is the shear modulus of the soil at the pile tip; and $T_b$ is the pile tip torque.

However, the torque load vector of the pile node cannot be expressed by Equation (35) in expansive soil. The pile tip is still separated from the soil when the vertical load is small, and torque is equal to zero. The pile tip exerts pressure on the soil when the vertical load is large, and the pile tip torque is not equal to zero. Therefore, the torsional load vector can be written as:

$$\{T_t''\} = \begin{cases} \left\{-2Th/\left(G_p J_p\right) 0 \dots 0 \dots 0\, 0\right\}^T & s_b(l) < 0 \\ \left\{-2Th/\left(G_p J_p\right) 0 \dots 0 \dots 0 - 3T_b h/\left(8G_L r_0^3\right)\right\}^T & s_b(l) \geq 0 \end{cases} \tag{37}$$

The Equation (35) is rewritten as:

$$[K_t']\{\theta\} = \{T_t''\} \tag{38}$$

Solving Equation (38) can obtain the pile node torsion angle vector:

$$\{\theta\} = [K_t']^{-1}\{T_t''\} \tag{39}$$

The torsion angle of a single pile under torque loading can be obtained by Equation (39), and the limit torque can be determined by plotting the torque–torsion angle curve at the pile top.

The process of solving Equation (39) using MATLAB is as follows:

(i)   Calculating the vertical displacement of the pile element according to Section 2.2, then the relative displacement $s(z)$ of the pile and soil can be obtained. Substituting it into the Equation (17), the vertical friction resistance $\tau_{sv}$ can be obtained.

(ii)  The circumferential ultimate friction of the pile $\tau_{tf}$ is achieved by solving Equation (32).

(iii) Assign the pile node torsion angle vector $\{\theta\}$ to zero vector, the relative torsional displacement of the pile–soil at depth z is calculated. Then, the torque stiffness matrix $[K_t']$ is obtained.

(iv)  Calculating displacement vector $\{\theta\}$ of pile node by Equation (39).

(v)   A new torque stiffness matrix $[K_t']^k$ is obtained using the new torsion angle vector $\{\theta\}^k$, and a new new torsion angle vector $\{\theta\}^{k+1}$ is obtained using the new vertical stiffness matrix.

(vi)  Taking the maximum value of elements in the $\left|\{\theta\}^{k+1} - \{\theta\}^k\right|$ as iterative control error. If the error is greater than the limit value, repeat (iii)~(vi) until the iterative error is less than the limit value.

## 3. Model Test

### 3.1. Model Pile

The model pile is made of aluminum alloy pipe and its outer diameter, inner diameter, elastic modulus, and buried depth are 0.025 m, 0.021 m, 69.7 GPa, and 0.7 m, respectively. Split the aluminum alloy pipe pile in half, and the BF350-3AA and BHF350-3HA strain gauges are attached to the inner surface of the model pile every 0.1 m so that the axial and shear strains are measured. Seven sets of measuring points were used to collect strain through the TST3822EN static strain tester. The surface of the pile was simulated by 502 glued with a layer of fine sand. A pipe hoop was installed at the pile top to prevent the model pile from being torn, and the pile end was sealed with nylon plugs.

### 3.2. Test Soil

The Guangxi expansive soil with 55% expansion rate and the expansive force measured by the test is 132 kPa. The liquid limit and the plastic limit are 53.2% and 31.2%, respectively. Other physical properties are shown in Table 1. The soil is dried and crushed to make a soil sample with a moisture content of 26%. The soil samples were artificially filled and compacted into layers. The thickness of each layer was 100 mm, and the total filling was 800 mm.

**Table 1.** Physical and mechanical properties of expansive soil.

| State | Density (g/cm$^3$) | Cohesion (kPa) | Internal Friction (°) |
|---|---|---|---|
| Before immersion | 1.92 | 39.6 | 18 |
| After immersion | 2.19 | 23.7 | 17.6 |

### 3.3. Loading and Measuring Device

As shown in Figure 6, the loading device is composed of a vertical loading device, a torque loading device, a water immersion device, and a model box. The vertical loading device is composed of a jack, a force sensor, and a reaction frame. A hinge is installed below the jack so that the pile top can rotate freely. The torque loading device consists of a loading frame and a pulley. The immersion device is shown in Figure 7, and the inlet pipe is connected to the external water tank and set at the bottom of the model box. The gravel, coarse sand, and fine sand are filled at the bottom of the model box, respectively. The perforated PVC pipe wrapped with geotextiles is installed in the coarse sand layer. Full saturation of the expansive soil can be achieved by making the height of the water level in the model box and the water tank consistent according to the siphon principle.

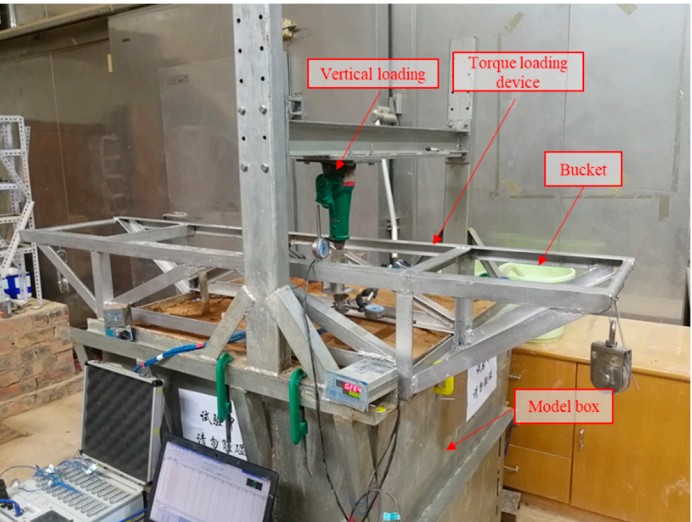

**Figure 6.** Test device diagram.

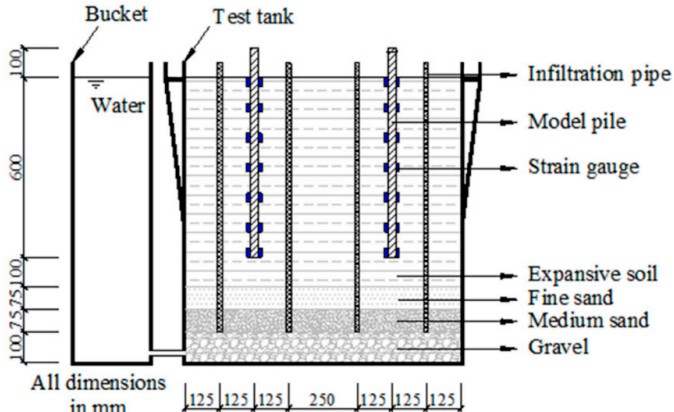

**Figure 7.** Schematic of immersion system.

The vertical force and vertical displacement are measured by LH-PT600 high-speed demonstrator and dial indicator, respectively. The TST3822EN static strain tester collects the shear strain, tensile and compressive strain of the pile.

### 3.4. Loading Scheme

The static load test can be started when there is no noticeable change in the dial indicator and strain gauge within 48 h. The test lasted for 15 days, and the ground heave was 0.011 m. The test is divided into two groups, as shown in Table 2. The first group is designed to test the ultimate vertical bearing capacity $V_u$. The second group pre-adds different vertical forces ($0$, $V_u/6$, $V_u/3$, $V_u/2$, $2V_u/3$, $5V_u/6$) to the pile top, and then applied the torque in stages until the limit torque $T_{ui}$ is reached. It should be noted that this device cannot accurately control the vertical force, and the torque can be applied when the vertical force is close to the data in Table 2.

**Table 2.** Test loading scheme.

| Group | Serial Number | Vertical Load ($V$) | | Torsional Load ($T$) | |
|---|---|---|---|---|---|
| | | Value | Ways of Loading | Value | Ways of Loading |
| 1 | $P_1$ | $V_u$ | Step loading | / | / |
| | $P_2$ | 0 | / | $T_{u1}$ | Step loading |
| | $P_3$ | $V_u/6$ | Constant | $T_{u2}$ | Step loading |
| 2 | $P_4$ | $V_u/3$ | Constant | $T_{u3}$ | Step loading |
| | $P_5$ | $V_u/2$ | Constant | $T_{u4}$ | Step loading |
| | $P_6$ | $2V_u/3$ | Constant | $T_{u5}$ | Step loading |

## 4. Validation

### 4.1. Solution to the Internal Force of the Pile after Immersion

Substituting the displacement of the pile $w_{swell}$ into Equations (4) and (5), the axial force of the pile can be obtained. In this section, the calculation results of the axial force on the pile will be validated.

Williams and Donaldson conducted long-term observations of a single concrete pile in expansive soil areas, as reported by Poulos and Davis [9]. The pile had a diameter of 9 inches (22.86 cm), a length of 34 feet (10.36 m), and was buried at a depth of 30 feet (9.144 m). The elastic modulus of the pile was 20 GPa, and the Poisson's ratio of the expansive soil was 0.3. The depth of the expansion influence was 5.18 m, and the ground heave was 10 mm. As shown in Figure 8, Xiao et al. [12], Poulos and Davis [9] utilized the aforementioned experimental measurements conducted by Williams and Donaldson to validate their calculation results. Xiao et al. [12] utilized the average stiffness method for calculation, resulting in a particular deviation from the experimental data and an overestimation of the axial force of the shallow foundation. The elastic calculation method adopted

by Poulos and Davis [9] could not fully reflect the complex elastoplastic relationship of the pile–soil interface, resulting in agreement in the upper half of the pile. However, the peak position and magnitude of the axial force still had significant errors compared to the tests. In contrast, the calculation method proposed in this paper is more consistent with the test data.

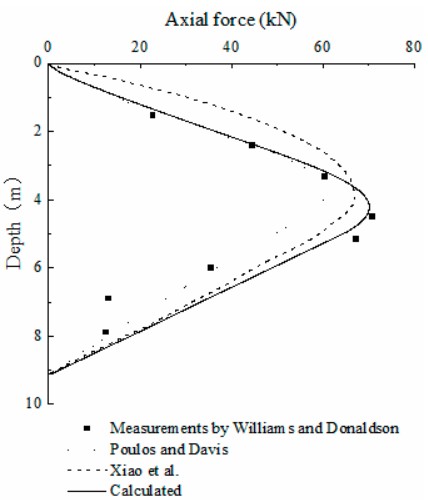

**Figure 8.** Comparison and validation of the axial force on the pile with the measurement results [9,12].

### 4.2. Nonlinear Solution of the Single Pile under Vertical Loading

A vertical load is applied after immersion. The relationship between the vertical force and the new displacement is shown in Figure 9. The load–displacement curve shows an inflection point at 150 N due to the reverse shear action at the pile–soil interface below the neutral point. The calculated results fit well with the experimental measurements in this study, which verifies the correctness of the proposed theory.

### 4.3. Calculation of Single Pile Bearing Capacity under V→T Loading Path

To verify the bearing capacity of the pile under the $V \rightarrow T$ loading path, the experimental results presented in the Section 3 were utilized. The torque–torsion angle curve of the pile top under a vertical load of 679 N is shown in Figure 10. It is evident from the figure that when the pile top torque is small, there may be some specific errors in the measurements due to a lack of accurate measurement methods during the test. In general, the theoretical calculation results are consistent with the experimental data.

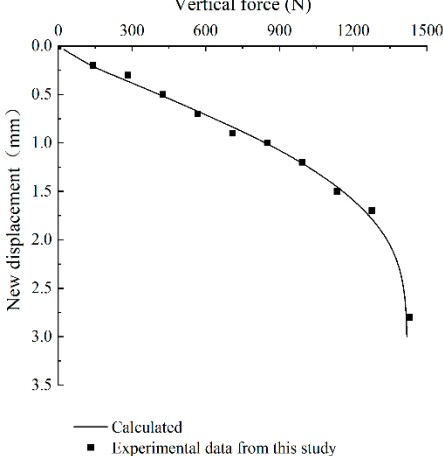

**Figure 9.** Comparison of Vertical force–new displacement curves at pile top and the experimental data of this study.

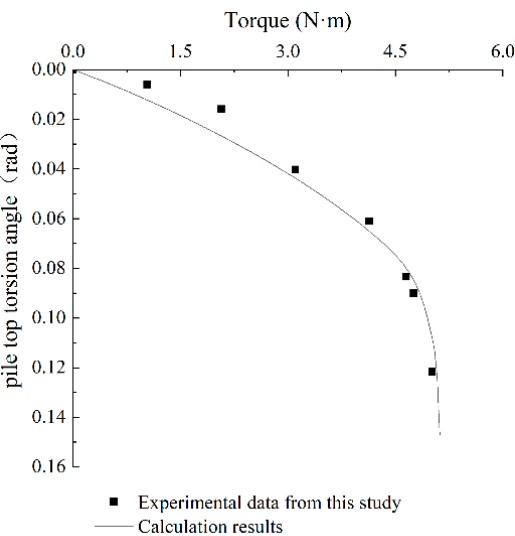

**Figure 10.** Comparison of the pile top Torque–torsion angle with the experimental data of this study under $V{\rightarrow}T$ loading path.

## 5. Parametric Study

### 5.1. Bearing Capacity Envelope

The bearing capacity envelopes of a single pile under the $V{\rightarrow}T$ loading path were plotted based on the experimental results in Section 3, as shown in Figure 11. The theoretical calculation results were consistent with the experimental data. However, the proposed method could not fully reflect the mechanical relationship between the pile–soil interface after immersion, leading to acceptable errors. Furthermore, the ultimate torque increased initially and then decreased with an increase in vertical load. When the vertical load reached $V_{\mathrm{u}}/3$, the torque bearing capacity reached a maximum. This was because, with an increase in vertical load, the neutral point gradually moved down, causing the vertical friction resistance below the neutral point to decrease initially and then increase in the opposite direction. This led to an initial increase and then a decrease in the space for exerting circumferential friction resistance. Therefore, proper pre-compression of the pile could enhance its torsion resistance in expansive soil foundations.

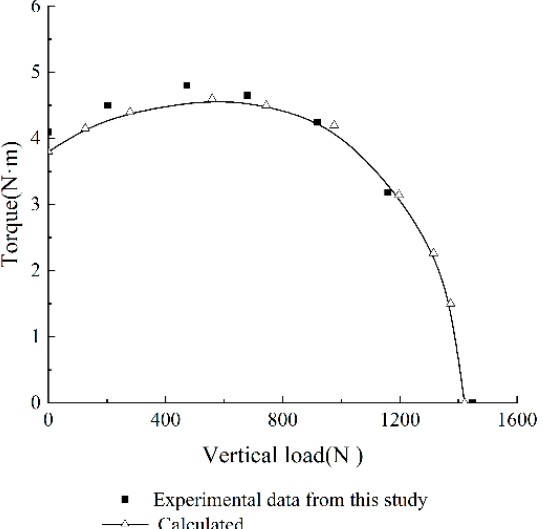

**Figure 11.** Comparison of single pile bearing capacity envelope with test data under $V{\rightarrow}T$ loading path.

## 5.2. Pile Deformation

By applying various vertical loads and torques to the pile top and using the procedures outlined in Sections 2.2 and 2.3, we were able to calculate the normalized vertical displacement and torsion angle of the pile. The normalized vertical displacement of the pile is shown in Figure 12a with applied vertical loads of 500 N, 800 N, 1000 N, and 1200 N to the pile top, and the normalized torsion angle of the pile is shown in Figure 12b with the torque of 1 N·m, 2 N·m, 3 N·m, and 4 N·m after an applied vertical load of 1000 N. It can be seen that deformation of the pile mainly occurs in the range of 0~0.6 L (shallow foundation). In addition, with the increase of torque and vertical load, the deformation ratio gradually increases and approaches the pile top. This is because the deformation of the pile develops from top to bottom. As the load increases, the resistance of the deep foundation gradually manifests. Therefore, strengthening the shallow foundation can effectively control pile deformation.

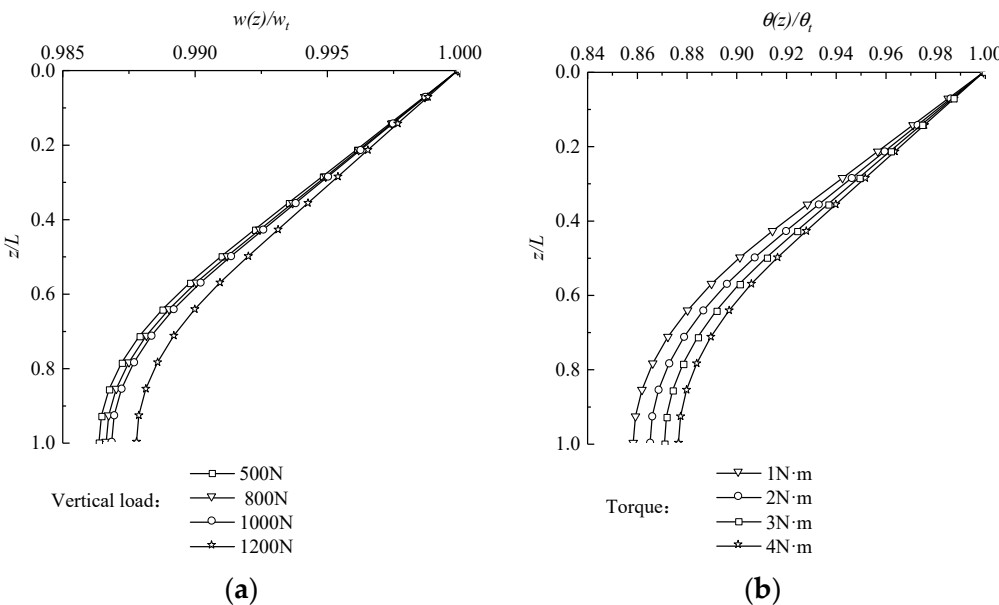

**Figure 12.** Normalized results under different loads. (**a**) Vertical displacement. (**b**) Torsion angle.

## 5.3. The Effect of Expansion Rate on Bearing Capacity

Using the calculation parameters from the model test conducted in this paper, we analyzed the impact of the expansion rate on the bearing capacity envelope using the calculation procedures outlined in Sections 2.2 and 2.3. The results are presented in Figure 13. The ground heave height corresponding to different expansion rates is 0.004~0.011 m. It can be seen from Figure 13 that with the increase of expansion rate, the bearing capacity of a single pile decreases. In addition, when the vertical load is less than $0.9V_u$, the limit torque decreases rapidly with the increase of the expansion rate. When the vertical load is greater than $0.9V_u$, the limit torque is gradually reached. The reason is that for a single loaded (vertical force or torque) pile, as the expansion rate increases, the ultimate torque will decrease significantly, while the vertical bearing capacity does not change significantly. When the vertical load is greater than $0.9V_u$, the effect of negative friction has been eliminated, the vertical friction with different expansion rates on the pile is close to each other, and the range of variation of circumferential frictional resistance becomes smaller. Thus, the bearing capacity envelope is relatively close. On the contrary, when the vertical load is less than $0.9V_u$, the negative friction effect on the pile has not been eliminated, and the range of variation of circumferential frictional resistance becomes larger. Therefore, the corresponding bearing capacity envelopes vary significantly.

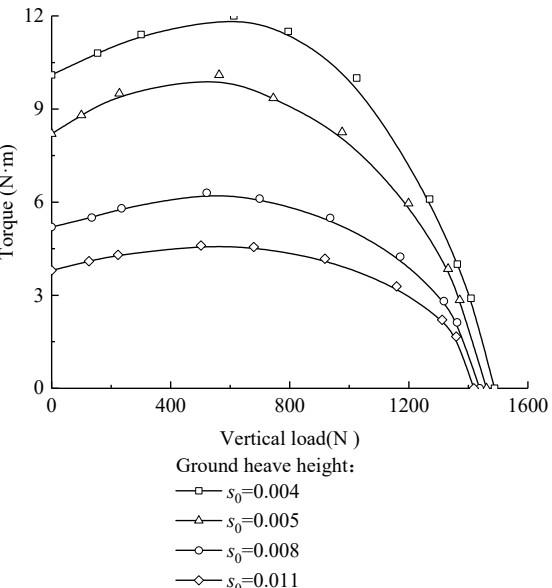

**Figure 13.** Effect of expansion ratio on the envelope of single pile bearing capacity.

## 6. Conclusions

(1) The pile displacement control equation is established based on the load transfer method. Considering the influence of the swelling deformation of expansive soil after immersion, the finite difference method is used to obtain the numerical solution of a single loaded pile. The load transfer function of the vertical positive and negative friction resistance of the single pile is established after immersion. Thus, the nonlinear solution of a single pile under vertical loading is obtained and the ultimate circumferential friction around the pile is corrected. The nonlinear solution of a single pile under the $V \rightarrow T$ loading path is gained and verified by the model test. In addition, the bearing capacity envelope was plotted.

(2) With the increase of the vertical load, the ultimate torque of the pile shows a trend of first increase and then decrease. Therefore, proper application of the vertical load can increase the torsion resistance of single piles.

(3) The deformation of the pile mainly occurs in the range of 0~0.6 L. Hence, strengthening the shallow foundation can effectively control the deformation.

(4) The bearing capacity of the pile decreases gradually with increasing expansion rate under the $V \rightarrow T$ loading path. When the vertical load is less than $0.9V_u$, the torsion resistance decreases significantly with increasing expansion rate. However, the reduction in torsion resistance becomes less pronounced when the vertical force is greater than $0.9V_u$.

**Author Contributions:** Z.W. conceptualized the research project and designed the study methodology. J.J. conducted the data collection and analysis, secured funding, and supervised the research team. S.W. contributed to the literature review and wrote the introduction and discussion sections. C.F. provided critical feedback on the manuscript and revised it for clarity. D.Y. formatted the paper. All authors have read and agreed to the published version of the manuscript.

**Funding:** This research was supported by the National Natural Science Foundation of China (Grant No. 52068004) and Key Research Projects of Guangxi Province (Grant No. AB19245018).

**Institutional Review Board Statement:** Not applicable.

**Informed Consent Statement:** Not applicable.

**Data Availability Statement:** Some or all data, models, or code generated or used during the study are available from the corresponding author by request.

**Conflicts of Interest:** The authors declare that they have no conflict of interest.

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
