# Peer review of "Analysis on Bearing Behavior of Single Pile under Combined Action of Vertical Load and Torque in Expansive Soil"

_applsci, doi:10.3390/app13074133_

Round 1

Reviewer 1 Report

The paper proposes  determining limit torque under the action of vertical force to analyze the bearing characteristics of curved beam bridge pile foundation under the combined action of vertical force and torque. The load transfer function at the pile-soil interface with positive and negative friction resistance is established. The nonlinear solution of a single pile under vertical load is achieved by the finite difference method. Then, the circumferential limit friction resistance is modified through the boundary element method, and a loop iteration program is compiled through MATLAB.

I see some merit in the paper and I would recommend publication after substatial revision. The major problem of the article is the lack of details about the numerical methods used in the analysis. For example, there isn't any information about the boundary element method, neither in the text nor in the reference. I looked other references of the same authors and in none of them I found details of the numerical formulation. I suggest major review of the article to clarify the employed numerical methods. Do you use a home-made boundary element code? If yes, you should give full details about the formulation of this code (boundary integral equations, fundamental solutions, degree of boundary elements, etc.). It cannot be a complete black box. If you have used a commercial code or even an open source boundary element code, you should cite the code and give details about your numerical example (how many boundary elements have you used?, what are the boundary conditions used in the boundary element model?, etc.). There isn't any reference to any boundary element book. 

About finite differences, there are little details, in my view, enough for those who know the method. For example, although there isn't details how you obtain equation (33) from (32), it is ok, it is a one-dimensional finite difference scheme that is present in many books about numerical method.  However, we cannot say the same about  the solving process of equation (36). The last step, "Eq. (36) can be solved by the iterative method of literature [7]", it is not enough. You must give details fo this iterative method. It is not clear to me how did you solve equation (36).

Other minor corrections:

- pag 2, line 86: "in this paper" instead of "in this pepper".

- The quality of Figures 1 and 3 is terrible. Please, redo them.

- Applied Sciences is an international journal. So, avoid citations in Chinese. For example, I got reference 17 but the paper is in Chinese. I could only understand the abstract, equations and pictures.

Author Response

Dear Editors and Reviewers:

Thank you for your letter and for the reviewers’ comments concerning our manuscript entitled “Analysis on bearing behavior of single pile under combined action of vertical load and torque in expansive soil” by Zhi Wang, Jie Jiang, Shunwei Wang, Chenzhi Fu, Di Yang. Those comments are all valuable and very helpful for revising and improving our paper. All the comments have been addressed and corresponding changes have been made to improve the manuscript. Point-by-point responses to the editors and reviewers are listed below:

  1. I see some merit in the paper and I would recommend publication after substatial revision. The major problem of the article is the lack of details about the numerical methods used in the analysis. For example, there isn't any information about the boundary element method, neither in the text nor in the reference. I looked other references of the same authors and in none of them I found details of the numerical formulation. I suggest major review of the article to clarify the employed numerical methods. Do you use a home-made boundary element code? If yes, you should give full details about the formulation of this code (boundary integral equations, fundamental solutions, degree of boundary elements, etc.). It cannot be a complete black box. If you have used a commercial code or even an open source boundary element code, you should cite the code and give details about your numerical example (how many boundary elements have you used?, what are the boundary conditions used in the boundary element model?, etc.). There isn't any reference to any boundary element book. 

Reply: Thank you for the valuable feedback from the experts. This article uses a finite difference method, and the description related to the boundary element method is inappropriate, so the description of the boundary element method has been removed from the manuscript.

  1. About finite differences, there are little details, in my view, enough for those who know the method. For example, although there isn't details how you obtain equation (33) from (32), it is ok, it is a one-dimensional finite difference scheme that is present in many books about numerical method. However, we cannot say the same about the solving process of equation (36). The last step, "Eq. (36) can be solved by the iterative method of literature [7]", it is not enough.  You must give details fo this iterative method.  It is not clear to me how did you solve equation (36).

Reply: Modifications have already been made in the original manuscript. A revised description of the iterative program is provided.

The formula (35) is rewritten as:

                                       (38)

Solving equation (38) can obtain the pile node torsion angle vector:

                                   (39)

The torsion angle of a single pile under the torque action can be obtained by formula (39), and the limit torque can be determined by plotting the torque-torsion angle curve at the pile top.

The process of solving equation (39) using MATLAB is as follows:

  • (i) Calculating the vertical displacement of the pile element according to section 2.2, then the relative displacement of the pile and soil can be obtained. Substituting it into the formula (17), the vertical friction resistance  can be obtained.
  • (ii) The circumferential ultimate friction of the pile body is achieved by solving equation (32).
  • (iii) Assign the pile node torsion angle vector to zero vector, the relative torsional displacement of the pile-soil at depth z is calculated. Then, the torque stiffness matrix  is obtained.
  • (iv) Calculating displacement vector of pile node by formula (39).
  • (v) A new torque stiffness matrix is obtained using the new torsion angle vector , and a new new torsion angle vector  is obtained using the new vertical stiffness matrix.
  • (vi) Taking the maximum value of elements in the as iterative control error. If the error is greater than the limit value, repeat (â…²)~(â…µ) until the iterative error is less than the limit value.

  1. pag 2, line 86: "in this paper" instead of "in this pepper".

Reply: "in this pepper" has been changed to "in this paper" and marked in red font.

  1. The quality of Figures 1 and 3 is terrible. Please, redo them.

Reply: We have made revisions to Figures 1 and 3 and have updated the original text accordingly. The updated versions of the figures are shown below:

Figure 1. Pile-soil interface model of expansive soil after immersion.

Figure 3. Load transfer curve of pile-soil interface under vertical load.

  1. Applied Sciences is an international journal. So, avoid citations in Chinese. For example, I got reference 17 but the paper is in Chinese. I could only understand the abstract, equations and pictures.

Reply: References to reference 17 have been removed, and all important references in the article are replaced in English.

All revised portions are marked in red in the revised manuscript!

Reviewer 2 Report

The manuscript presents an interesting analysis regarding bearing capacity of single pile under the loading path of V→T, in expansive soil – an issue which is rather present in case of curved beam bridge construction. New calculation method has been proposed and verified.

The study has reasonable significance, its findings may be important for and proposals useful in the future design of piles in expansive soils. The paper topic and content are appropriate for its publication in special issue of journal Applied Sciences - Urban Underground Engineering: Excavation, Monitoring, and Control.

The article is well written, in general. It is properly organised, not too demanding for reading and understanding, exhibiting good command of English. A few improvements regarding additional information or more precise clarifications would be welcome. Technical editing should be improved, primarily in photo/drawing illustrations. A proper range of literature was referenced, although mainly older publications.

I recommend accepting the paper, after minor revision.

Comments:

1/ The literature should be referred to in unified style throughout the article. All the references with more than one author should be mentioned in the article text in the same way – either only by the first author or by all authors (x and y; x at al etc), depending on journal’s rules for technical editing.

2/ Text editing corrections are needed in lines 99, 107, 281, 383… where some parts of text are (probably unintentionally) moved into superscript position.

3/ Technical quality of illustrations (primarily diagrams – Figures 1, 3, but also other drawings – Figure 5a…) should be significantly improved. It seems like lines are duplicated (thicker line over thinner line), but not perfectly overlapping. Hence, final appearance of drawing is rather messy.

4/ Figures 2 and 4 has the same title. Some difference should be noted in their tittle.

5/ What does “repeating steps 2” (line 249) refer to?

6/ While subscript “sv” is used in symbol for vertical friction resistance in text (the first paragraph in Section 2.3.1 and onward), the same is denoted with subscript “v” in Figure 5. Symbols should be adjusted.

7/ The article would benefit from more detailed illustration of test setup. Maybe Figure 6 may contain few photos with details described in the Section 3.3, instead of only one photo, presenting general overview.

8/ Please, check the last sentence of the first paragraph in the Section 4.1 – “…the axial force of the pile is be verified.” (line 376).

9/ Please, check referencing in the second paragraph in the Section 4.1 – reference [16].

10/ It should be clearly stated in the Section 4.1 (in text and particularly in the title and legend of Figure 8) which “measured” data are used – the old data from observations of Williams and Donaldson?

11/ It should be clearly stated in the Section 4.2 (in text and particularly in the title and legend of Figure 9) which “experimental data” are used – the data from tests presented in the article, in the Section 3?

12/ It should be clearly stated in the Section 4.3 (in text and particularly in the title and legend of Figure 10) which “experimental data” are used – the data from tests presented in the article, in the Section 3?

13/ An explanation of selected value of vertical load of 679 N in the Section 4.3 would be welcome.

14/ It should be clearly stated in the Section 5.1 (in text and particularly in the title and legend of Figure 11) which “experimental data” are used – the data from tests presented in the article, in the Section 3?

15/ It should be clearly stated in the Section 5.2 if experimental (from Section 3) or calculation (by means of procedure from Section 2) results are presented in Figure 12.

16/ Further explanation/clarification of the “above model test” mentioned in the Section 5.3 would be welcome. It should be clearly stated in the Section 5.3 if experimental (from Section 3) or calculation (by means of procedure from Section 2) results are presented in Figure 13.

17/ Typing correction in line 479 is necessary – “smaller.when”.

18/ A statement from Abstract “a loop iteration program is compiled through MATLAB” has not been addressed in the paper. A short comment in the article body text would be welcome.

19/ The part “Authors contribution” is missing. Necessity of its presence in the article should be discussed with the Technical Editor.

Author Response

Dear Editors and Reviewers:

Thank you for your letter and for the reviewers’ comments concerning our manuscript entitled “Analysis on bearing behavior of single pile under combined action of vertical load and torque in expansive soil” by Zhi Wang, Jie Jiang, Shunwei Wang, Chenzhi Fu, Di Yang. Those comments are all valuable and very helpful for revising and improving our paper. All the comments have been addressed and corresponding changes have been made to improve the manuscript. Point-by-point responses to the editors and reviewers are listed below:

  1. The literature should be referred to in unified style throughout the article. All the references with more than one author should be mentioned in the article text in the same way – either only by the first author or by all authors (x and y; x at al etc), depending on journal’s rules for technical editing.

Reply: The format of references in the text has been carefully checked, and inappropriate citations have been corrected.

  1. Text editing corrections are needed in lines 99, 107, 281, 383… where some parts of text are (probably unintentionally) moved into superscript position.

Reply: The format of (in lines 99), [19] (in lines 107), [17] (in lines 281), [12] (in lines 383) has been modified, and corrected other similar errors.

  1. Technical quality of illustrations (primarily diagrams – Figures 1, 3, but also other drawings – Figure 5a…) should be significantly improved. It seems like lines are duplicated (thicker line over thinner line), but not perfectly overlapping. Hence, final appearance of drawing is rather messy.

Reply: The figure has been redrawn and updated in the manuscript. The specific results are as follows:

Figure 1. Pile-soil interface model of expansive soil after immersion.

Figure 3. Load transfer curve of pile-soil interface under vertical load.

Figure 5. (a) Discrete model of pile surface (b) Shear element model.

  1. Figures 2 and 4 has the same title. Some difference should be noted in their tittle.

Reply: “Figure 2 Schematic of difference discrete of a single pile.” has been changed to “Figure 2 Schematic diagram of differential dispersion of single pile after immersion”, and “Figure 4. Schematic of difference discrete of a single pile” has been changed to “Schematic diagram of differential dispersion of single pile under vertical load after water immersion”.

  1. What does “repeating steps 2” (line 249) refer to?

Reply: “repeating steps â‘¡” (line 249) has been changed to “repeating steps (â…±) (line 251)”.

  1. While subscript “sv” is used in symbol for vertical friction resistance in text (the first paragraph in Section 2.3.1 and onward), the same is denoted with subscript “v” in Figure 5. Symbols should be adjusted.

Reply: Modifications have been made to the notation in Figure 5, which can be seen in Question 3.

  1. The article would benefit from more detailed illustration of test setup. Maybe Figure 6 may contain few photos with details described in the Section 3.3, instead of only one photo, presenting general overview.

Reply: The corresponding loading device in Figure 6 has been marked to make it easier for readers to understand, as shown in the figure below.

Figure 6 Test device diagram

  1. Please, check the last sentence of the first paragraph in the Section 4.1 – “…the axial force of the pile is beverified.” (line 376).

Reply: “the axial force of the pile is be verified” has been changed to “the calculation results of the axial force on the pile will be validated.”

  1. Please, check referencing in the second paragraph in the Section 4.1 – reference [16].

Reply: We have made the following modifications to the cited references: Williams and Donaldson conducted long-term observations of a single concrete pile in expansive soil areas, as reported by Poulos and Davis [9]. (Change [16] to [9])

  1. It should be clearly stated in the Section 4.1 (in text and particularly in the title and legend of Figure 8) which “measured” data are used – the old data from observations of Williams and Donaldson?

Reply: A description of the experimental data has been added to the manuscript and figures, and this passage has been polished as follows:

Williams and Donaldson conducted long-term observations of a single concrete pile in expansive soil areas, as reported by Poulos and Davis [9]. The pile had a diameter of 9 inches (22.86 cm), a length of 34 feet (10.36 m), and was buried at a depth of 30 feet (9.144 m). The elastic modulus of the pile was 20 GPa, and the Poisson's ratio of the expansive soil was 0.3. The depth of the expansion influence was 5.18 m, and the ground heave was 10 mm. As shown in Figure 8, Xiao et al. [12] and Poulos and Davis [9] utilized the aforementioned experimental measurements conducted by Williams and Donaldson to validate their calculation results. Xiao et al. [12] utilized the average stiffness method for calculation, resulting in a particular deviation from the experimental data and an overestimation of the axial force of the shallow foundation. The elastic calculation method adopted by Poulos and Davis [9] could not fully reflect the complex elastoplasticity relationship due to the elasto-plastic load-displacement relationship of the pile-soil interface, resulting in only the upper half of the pile being consistent. However, the peak position and magnitude of the axial force still had significant errors compared to the tests. In contrast, the calculation method proposed in this paper is more consistent with the test data.

Figure 8. Comparison and validation of the axial force on the pile with the measurement results of Williams and Donaldson[9].

  1. It should be clearly stated in the Section 4.2 (in text and particularly in the title and legend of Figure 9) which “experimental data” are used – the data from tests presented in the article, in the Section 3?

Reply: A description of the experimental data has been added to the manuscript and figures as follows:

The calculated results fit well with the experimental measurements in this study, which verifies the correctness of the proposed theory.

Figure 9. Comparison of Vertical force-new displacement curves at pile top and the experimental data of this study.

  1. It should be clearly stated in the Section 4.3 (in text and particularly in the title and legend of Figure 10) which “experimental data” are used – the data from tests presented in the article, in the Section 3?

Reply: A description of the experimental data has been added to the manuscript and figures, and this passage has been polished as follows:

To verify the bearing capacity of the pile under the V→T loading path, the experimental results presented in the section 3 were utilized. The torque-torsion angle curve of the pile top under a vertical load of 679 N is shown in Figure 10. It is evident from the figure that when the pile top torque is small, there may be some specific errors in the measurements due to a lack of accurate measurement methods during the test. In general, the theoretical calculation results are consistent with the experimental data.

Figure 10. Comparison of the pile top Torque-torsion angle with experimental data of this study under V→T loading path.

  1. An explanation of selected value of vertical load of 679 N in the Section 4.3 would be welcome.

Reply: In section 3.4, a loading scheme is listed. However, due to the displacement loading method used in this experiment, it is not possible to precisely control the vertical load and torque values. Only slow loading can be performed, and experimental data should be recorded when the loading data is close to Table 2. In this experiment, the vertical load of 679N is randomly generated, and we cannot control it precisely.

  1. It should be clearly stated in the Section 5.1 (in text and particularly in the title and legend of Figure 11) which “experimental data” are used – the data from tests presented in the article, in the Section 3?

Reply: A description of the experimental data has been added to the manuscript and figures, and this passage has been polished as follows:

The bearing capacity envelopes of a single pile under the V→T loading path were plotted based on the experimental results in Section 3, as shown in Figure 11. The theoretical calculation results were consistent with the experimental data, but the proposed method could not fully reflect the mechanical relationship between the pile-soil interface after immersion, leading to acceptable errors. Furthermore, the ultimate torque increased initially and then decreased with an increase in vertical load. When the vertical load reached Vu/3, the torque bearing capacity reached a maximum. This was because, with an increase in vertical load, the neutral point gradually moved down, causing the vertical friction resistance below the neutral point to decrease initially and then increase in the opposite direction. This led to an initial increase and then a decrease in the space for exerting circumferential friction resistance. Therefore, proper pre-compression of the pile could enhance its torsion resistance in expansive soil foundation.

Fig. 11 Comparison of single pile bearing capacity envelope with test data under V→T loading path.

15.It should be clearly stated in the Section 5.2 if experimental (from Section 3) or calculation (by means of procedure from Section 2) results are presented in Figure 12.

Reply: A description of the calculation procedure has been added to the manuscript, as follows:

By applying various vertical loads and torques to the pile top and using the pro-cedures outlined in Section 2.2 and Section 2.3, we were able to calculate the normalized vertical displacement and torsion angle of the pile body.

16.Further explanation/clarification of the “above model test” mentioned in the Section 5.3 would be welcome. It should be clearly stated in the Section 5.3 if experimental (from Section 3) or calculation (by means of procedure from Section 2) results are presented in Figure 13.

Reply: A description of the calculation procedure has been added to the manuscript, as follows:

Using the calculation parameters from the model test conducted in this paper, we analyzed the impact of the expansion rate on the bearing capacity envelope using the calculation procedures outlined in Sections 2.2 and 2.3. The results are presented in Figure 13.

17.Typing correction in line 479 is necessary – “smaller.when”.

Reply: The following modifications have been made in the manuscript,:

The bearing capacity of the pile decreases gradually with increasing expansion rate under the V→T loading path. When the vertical load is less than 0.9Vu, the torsion resistance decreases significantly with increasing expansion rate. However, the reduction in torsion resistance becomes less pronounced when the vertical force is greater than 0.9Vu.

18/ A statement from Abstract “a loop iteration program is compiled through MATLAB” has not been addressed in the paper. A short comment in the article body text would be welcome.

Reply: The description of the MATLAB iterative program has been added to the paper., which is as follows:

The iterative process for solving equation (14) using Matlab is as follows (line 177):

The iterative process for calculating the displacement of a vertically loaded pile considering the influence of immersion in expansive soil using MATLAB is as follows (line 253):

The process of solving equation (39) using MATLAB is as follows (line 323):

19.The part “Authors contribution” is missing. Necessity of its presence in the article should be discussed with the Technical Editor.

Reply: The author's contribution to this article has been added to the manuscript. Here are the contributions of five authors to this article:

Zhi Wang conceptualized the research project and designed the study methodology. Jie Jiang conducted the data collection and analysis, secured funding, and supervised the research team. Shun-wei Wang contributed to the literature review and wrote the introduction and discussion sections. Chen-zhi FU provided critical feedback on the manuscript and revised it for clarity. Di Yang formatted the paper.

All revised portions are marked in red in the revised manuscript!

Reviewer 3 Report

The English language of the article is not sufficient and needs improvement.

Figures 1 and 3 are illegible: the lines are too thick and there are pieces of text and symbols covered by a line

Figure 5 is also illegible: why there are two types of arrows on each other?

Maybe some short description of what exactly is expansive soil could appear in the introduction

l.44 Why word "Combined" starts with capital letter?

l. 105-108 In literature, the fi' is named mainly as an effective internal friction angle

l. 107 why [19] is in superscript?

l. 119, what is h0

l. 148 and l. 150 should be written as (for example) bullet points, because it could make the paragraph more readable

l. 158 and l. 160 w_swell and Fv' should be written in new lines as formulas

l. 170 space needed after node

l. 204 and l. 207 also should be written as bullet points

Author Response

Dear Editors and Reviewers:

Thank you for your letter and for the reviewers’ comments concerning our manuscript entitled “Analysis on bearing behavior of single pile under combined action of vertical load and torque in expansive soil” by Zhi Wang, Jie Jiang, Shunwei Wang, Chenzhi Fu, Di Yang. Those comments are all valuable and very helpful for revising and improving our paper. All the comments have been addressed and corresponding changes have been made to improve the manuscript. Point-by-point responses to the editors and reviewers are listed below:

  1. The English language of the article is not sufficient and needs improvement.

Reply: Thank you for your feedback, and we have carefully considered your comment regarding the insufficient English language of our article. We have reviewed and revised the English expression of the article to enhance its quality and readability.

  1. Figures 1 and 3 are illegible: the lines are too thick and there are pieces of text and symbols covered by a line

Reply: We have modified and redrawn Figures 1 and 3 and updated the original text accordingly. The updated figures are as follows:

Figure 1. Pile-soil interface model of expansive soil after immersion.

Figure 3. Load transfer curve of pile-soil interface under vertical load.

  1. Figure 5 is also illegible: why there are two types of arrows on each other?

Reply: The different arrows in Figure 5 represent different forces, and using different types of arrows can make it clearer which force each arrow represents. The figure has been redrawn and updated in the manuscript accordingly. The updated figures are as follows:

Figure 5. (a) Discrete model of pile surface (b) Shear element model.

  1. Maybe some short description of what exactly is expansive soil could appear in the introduction

Reply: A description of expansive soil has been added to the introduction section of the original manuscript, as follows: Expansive soil is a highly plastic and cohesive soil that typically exists in a hard and strong state in its natural state. It is sensitive to moisture and can swell when it absorbs water, resulting in significant deformation when it is in a free state. If deformation is restricted, it can generate expansive force. Conversely, when it loses water, its volume can shrink, leading to cracks and a decrease in strength. Its high degree of expansiveness can cause building structures to tilt, deform, and undergo uneven settlement, presenting significant engineering problems.

  1. L.44 Why word "Combined" starts with capital letter?

Reply: The word "Combined" has been changed to "combined" in the manuscript.

  1. L.105-108 In literature, the fi' is named mainly as an effective internal friction angle

Reply: The meaning of the letters has already been modified in the manuscript.  is the effective internal friction angle of the pile-soil interface, and  is the internal friction angle of the soil.

  1. L.107 why [19] is in superscript?

Reply: The superscript for 19 has been removed in the manuscript.

  1. L.119, what is h0

Reply:  is the depth at which expansion occurs, and soil below the depth of h will not experience expansion.

  1. L.148 and L. 150 should be written as (for example) bullet points, because it could make the paragraph more readable

Reply: Bullet points have been added in lines L.148 and L.150.

  1. L.158 and L. 160 w_swell and Fv' should be written in new lines as formulas

Reply:  and  have been presented in the manuscript in the form of equations.

11.L. 170 space needed after node

Reply: A space has been added after the node.

  1. 204 and l. 207 also should be written as bullet points

Reply: The bullet points have been added to the corresponding positions in the original manuscript at line 204 and 207.

All revised portions are marked in red in the revised manuscript!

Round 2

Reviewer 1 Report

Congratulations for the revision work. Now I consider that the article is ready to be published.